# Malnutrition Aggravates Alterations Observed in the Gut Structure and Immune Response of Mice Infected with *Leishmania infantum*

**DOI:** 10.3390/microorganisms9061270

**Published:** 2021-06-11

**Authors:** Felipe Gaitán-Albarracín, Monica Losada-Barragán, Nathalia Pinho, Renata Azevedo, Jonathan Durães, Juan Sebastián Arcila-Barrera, Rodrigo C. Menezes, Fernanda N. Morgado, Vinicius de Frias Carvalho, Adriana Umaña-Pérez, Patricia Cuervo

**Affiliations:** 1Laboratório de Pesquisa em Leishmanioses, Instituto Oswaldo Cruz, Fiocruz, 21040-360 Rio de Janeiro, Brazil; faga_1986@hotmail.com (F.G.-A.); nathps@ioc.fiocruz.br (N.P.); r.a.n@hotmail.com.br (R.A.); jonathan.duraes75@gmail.com (J.D.); morgado@ioc.fiocruz.br (F.N.M.); 2Grupo de Investigación en Hormonas, Departamento de Química, Facultad de Ciencias, Universidad Nacional de Colombia, Sede Bogotá, 111321 Bogotá, Colombia; jsarcilab@unal.edu.co; 3Grupo de Investigación en Biología Celular y Funcional e Ingeniería de Biomoléculas, Universidad Antonio Nariño, 111511 Bogotá, Colombia; moni.losada@gmail.com; 4Instituto Nacional de Infectologia Evandro Chagas, Fiocruz, 21040-360 Rio de Janeiro, Brazil; rodrigo.menezes@ini.fiocruz.br; 5Laboratório de Inflamação, Instituto Oswaldo Cruz, Fiocruz, 21040-360 Rio de Janeiro, Brazil; vfrias@ioc.fiocruz.br

**Keywords:** *Leishmania infantum*, visceral leishmaniasis, malnutrition, gut, duodenum, inflammation, IgA

## Abstract

Malnutrition is a risk factor for developing visceral leishmaniasis and its severe forms. Our group demonstrated that malnourished animals infected with *Leishmania infantum* had severe atrophies in lymphoid organs and T cell subpopulations as well as altered levels of thymic and splenic chemotactic factors, all of which resulted in dysfunctional lymphoid microenvironments that promoted parasite proliferation. Here, we hypothesize that malnutrition preceding parasite infection leads to structural and immunological changes in the gut mucosae, resulting in a failure in the immune response sensed in the intestine. To evaluate this, we analyzed the immunopathological events resulting from protein malnutrition in the guts of BALB/c mice infected with *L. infantum*. We observed lymphocytic/lymphoplasmacytic inflammatory infiltrates and lymphoid hyperplasia in the duodenum of well-nourished-infected mice; such alterations were worsened when malnutrition preceded infection. Parasite infection induced a significant increase of duodenal immunoglobulin A (IgA) of well-nourished animals, but those levels were significantly decreased in malnourished-infected mice. In addition, increased levels of Th17-related cytokines in duodenums of malnourished animals supported local inflammation. Together, our results suggest that the gut plays a potential role in responses to *L. infantum* infection—and that such responses are impaired in malnourished individuals.

## 1. Introduction

Malnutrition is a persistent concern in low-income countries and represents a high health burden on a global scale [1]. At early life stages, low nutrient intake can impact a child’s development and successful entrance into society, perpetuating the cycle of poverty and malnutrition. Visceral leishmaniasis (VL) is a neglected disease that frequently afflicts malnourished populations. Indeed, malnutrition is a risk factor for developing VL and its severe forms. The disease is caused by *Leishmania infantum* or *L*. *donovani* parasites, which canonically infect the spleen, liver, bone marrow, and lymph nodes, causing fever, hepatosplenomegaly, and loss of weight. However, in immunocompromised individuals coinfected with HIV, the parasite can be potentially detected in alternative locations, e.g., the gastrointestinal, respiratory, and cerebrovascular systems, among others [2,3]. Immunosuppression is one of the distinctive features observed in malnourished individuals, increasing their susceptibility to VL and limiting its resolution [4,5,6,7]. We previously demonstrated the deleterious effects of malnutrition on lymphoid organs in mice infected with *L. infantum* [8,9]. Malnutrition modifies thymic and splenic microenvironments, alters T cell subpopulations, and induces decreased levels of chemotactic molecules involved in the migration of T cells [8,9,10]. Additionally, malnutrition interferes with the body’s ability to control the proliferation of the parasite in the spleen, diminishing local and systemic proinflammatory responses and leading to an impaired cell-mediated immune response in infected animals. This indicates that a precondition of protein malnutrition affects the protective response against *L. infantum* [9,10,11].

The intestine is a crucial organ for nutritional sensing, microbiome homeostasis, and local immune response. The gastrointestinal lymphoid tissue encompasses around half of the body’s lymphocytes and plays an essential role in local and systemic immunity [12]. The presence of parasites invading the gut is frequently observed in dogs with chronic VL. The dissemination of the parasite in all segments of the intestine causes structural and functional changes that presumably increase the animal’s systemic deterioration [12]. Gastrointestinal symptoms have also been described in VL patients [13], including deficiency in vitamin A absorption, diarrhea [14], low levels of cytokines in the duodenum, and mild villous intestinal atrophy [15]. However, the nature of crosstalk through the three-way axis—malnutrition-*Leishmania*-gut—is unknown.

Here, we hypothesize that protein malnutrition and *L. infantum* infection leads to structural and immunological changes in the intestinal tissue, mediated by disorganization in its microarchitecture and dysregulation of cytokine abundance, leading to an altered immune response in the gut when malnourished animals are infected with *L. infantum*. To evaluate this, we analyzed the immunopathological events resulting from protein malnutrition in the gut of BALB/c mice infected with *L. infantum*.

## 2. Materials and Methods

### 2.1. Ethics Statement

This study was carried out following the recommendations in the Guide for the Care and Use of Laboratory Animals of the National Institutes of Health—Eighth Edition. All animal procedures were approved by the Instituto Oswaldo Cruz Animal Care and Use Committee (License CEUA-IOC, No. 038/2018). The *L. infantum* strain (MCAN/BR/2000/CNV-FEROZ) used in this study was provided by the collection of *Leishmania* of the Instituto Oswaldo Cruz, Rio de Janeiro (CLIOC http://clioc.fiocruz.br/). In accordance with the Brazilian Law of Biodiversity, this study was registered at SisGen (AA2236F).

### 2.2. Parasite Culture

Parasites were cultured at 25 °C in Schneider’s Insect Medium (Sigma-Aldrich, St. Louis, MO, USA) containing 10% fetal bovine serum (FBS) and were collected at the stationary phase by centrifugation at 1800× *g* for 5 min. The parasites were then washed twice in PBS, pH 7.2.

### 2.3. Mice, Feeding Protocol and Experimental Infection

The BALB/c model of malnutrition was developed as previously described [8,9]. Briefly, male BALB/c mice of three weeks of age (*n* = 48) received a standard diet with a 14% protein content (MP Biomedicals, Santa Ana, CA, USA). The mice were maintained on 12/12 h of light/dark cycle. After one week, the animals were randomly divided into two groups: 24 animals were fed a 14% protein diet and 24 animals were fed a 4% protein diet (MP Biomedicals, Santa Ana, CA, USA). Both diets were isocaloric, and each one provided 3.7 Kcal/g. The animals had free access to water and food. After 7 days of diet, each group of animals was divided into two subgroups, in which one was infected with *L. infantum* (1 × 10^7^ parasites) intravenously (caudal vein), while the other received saline solution. Thus, four experimental groups were obtained: animals fed with 14% protein (Control Protein, CP) (*n* = 12), animals fed with 4% protein (Low Protein, LP) (*n* = 12), animals fed with 14% protein and infected with *L. infantum* (CPi) (*n* = 12), and animals fed with 4% protein and infected (LPi) (*n* = 12). The diets were maintained after infection. The mice were monitored daily during infection and body weight was registered every three days. The animals were euthanized at 14 days after infection. Blood was collected by cardiac puncture and sera were separated and stored at −30 °C. The spleen was removed, weighed, and subsequently conserved for nucleic acid extraction. The intestine was sectioned in order to obtain the duodenum, jejunum, ileum, and colon; each tissue was gently washed with PBS to collect the luminal content. The luminal content from each tissue was further filtered with a 100 µm membrane to separate insoluble particles, centrifuged at 5000× *g*, and the supernatant (hereafter called luminal fluid) and pellet were stored at −80 °C. A fragment of each (duodenum, jejunum, ileum, or colon) was fixed in 10% buffered formalin for further histopathological analysis and immunohistochemistry (IHC) assay to detect amastigote forms of *Leishmania*—or frozen for gene expression analysis. For histopathology and IHC, the intestine of 6 mice per group were examined.

### 2.4. Hormone and Cytokine Levels

The serum levels of total IGF-1 and leptin were measured by ELISA assays (Abcam, Cambridge, UK) according to the manufacturer’s procedures. Likewise, the levels of corticosterone in serum were measured by radioimmunoassay using a gamma counter (ICN Isomedic 4/600) following the guidelines of the manufacturer (MP Biomedicals, Santa Ana, CA, USA). The presence of IL-7, IL-12, IL-13, IL-17A, IL-21, IL-22, IL-23, CCL5, CXCL9, CXCL10, TNFα, IFNγ and TGFβ in the luminal fluid of the duodenum and colon was analyzed using a multiplex immunoassay based on fluorescence-encoded beads according to the manufacturer’s instructions (BioLegend, San Diego, CA, USA). The acquisition was performed in a BD FACSCanto™ II flow cytometer (BD Biosciences, San Jose, CA, USA). Off-line analysis was performed with LEGENDplex™ (BioLegend, San Diego, CA, USA) data analysis software (version 8.0), and results were expressed as the mean of the reporter fluorescence intensity PE (MFI) as a function of concentration (pg/mL). Each assay was performed with two technical replicates.

### 2.5. Positivity of Infection and Parasite Load

The presence of *L. infantum* in the spleen and intestine of the infected animals was assessed by real-time quantitative PCR (qPCR) following the previously described protocols [16]. DNA was extracted from each organ using a commercial kit (Wizard Genomic DNA Purification System, Promega, São Paulo, Brazil) and dissolved in 100 µL of Tris-EDTA buffer. Parasite load was calculated using the Ubiquitin C (UBC) gene (Appendix A) as endogenous control, followed by normalization to cell number (19). The amplified target (kDNA) was visualized on a 1.5% agarose gel containing SYBR safe to confirm the positivity of all samples. 

### 2.6. Histopathological Analysis

Samples of duodenum, jejunum, ileum, and colon that were fixed in 10% buffered formalin were routinely processed for paraffin embedding. Paraffin blocks of these samples were cut in 5 µm thick sections, then stained with hematoxylin and eosin (H&E) [17] and analyzed under an optical microscope (Nikon Eclipse E400, Tokyo, Japan). Histological alterations in the morphology of the intestinal villi and intestinal crypts and the presence of inflammatory infiltrate in the different cell layers of the epithelium, lamina propria, Peyer’s patches, muscular layer of the mucosa, and submucosa were evaluated. The alterations were qualitatively described in the experimental groups and—in order to obtain a quantitative dimension of them—a score was assigned to each alteration (cellular and structural) according to its presence or absence and its severity degree (Appendix A). The intensity of inflammatory infiltration was classified as absent or mild (absent cellular infiltrate or mild and dispersed foci) and moderate to severe (dense and diffuse cellular infiltrate). The inflammatory infiltration was classified as transmural when extending through the entire thickness of the gut wall (lamina propria, submucosa and muscularis externa layers). All images were processed using the program ImageJ (NIH, Bethesda, MD, USA).

### 2.7. Immunohistochemistry for Detection of Leishmania

For detection of amastigote forms of *Leishmania* spp., gut tissues were submitted to consecutive steps of deparaffinization, rehydration, blocking of endogenous peroxidase, antigen retrieval, blockade of nonspecific protein binding, and incubation with polyclonal rabbit anti-*Leishmania* serum diluted 1:500 [18]. Parasite detection was made using a polymer-based detection system (HiDef Detection HRP^TM^ Polymer System, Cell Marque, Rocklin, CA, USA) according to manufacturer recommendations. Histological sections of organs intensely parasitized with amastigote forms of *Leishmania* were incubated with nonimmune homologous serum as negative control and with polyclonal rabbit anti-*Leishmania* serum as a positive control.

### 2.8. Gene Expression Analysis

Total RNA from duodenum and colon was extracted using the TRIzol^TM^ reagent according to the manufacturer’s instructions (Life Technologies, Carlsbad, CA, USA). The RNA was quantified with a Nanodrop ND-1000 spectrophotometer (Thermo Fisher Scientific, Waltham, MA, USA), and cDNA was synthesized from 1 μg of total RNA using the SuperScript III reverse transcription system (Invitrogen, Waltham, MA, USA). Real-time quantitative PCR was performed in technical duplicate using SYBR^®^ Green PCR Master Mix and the ViiA7 equipment (Applied Biosystems, Foster City, CA, USA), according to the manufacturer’s protocols. The analyzed transcripts corresponded to the coding genes for *il-10*, *il-12*, *ifn-γ*, *tnf-α*, *tgf-β*, *il-17a*, and the reference genes *hprt*, *atp-β5*, and *cyc-1* (Appendix A). PCR conditions and gene expression analyses were performed as previously described [8,9]. Data are shown as normalized relationships between the expression of the target gene and the geometric mean of the three reference genes. The qPCR analysis was carried out following the MIQE guidelines.

### 2.9. Quantification of Local and Systemic Soluble Immunoglobulin A (IgA) Levels

Systemic immunoglobulin A (IgA) levels were determined in the serum using an ELISA assay according to the manufacturer’s instructions (Invitrogen, Waltham, MA, USA). Local levels of secreted IgA were quantified in the luminal fluid obtained from each intestinal region using the same ELISA assay.

### 2.10. Statistical Analysis

Statistical analysis was performed using GraphPad Prism 8.0 software. Two-way analysis of variance (ANOVA) and Tukey’s *post-hoc* tests were used to analyze differences among treatments. Two-way ANOVA allowed us to examine the interaction between the two independent variables of our study: malnutrition and infection. Thus, significant differences (*p* < 0.05) due to diet, infection, or interaction between both variables are denoted with a, b, or c, respectively, at each figure. The Student *t*-test was used to analyze differences in body weight due to diet treatments (CP or LP) before infection and to analyze differences in parasite load between CPi and LPi animals.

## 3. Results

### 3.1. Whereas Systemic Levels of IGF-1 and Leptin Are Significantly Decreased and Corticosterone and IgA Are Significantly Increased in Malnourished BALB/c Mice

To corroborate the nutritional status of our model, we monitored the body weight of the animals every third day. In addition, at the end of the experiment, we measured markers of malnutrition and stress, including IGF-1, leptin, corticosterone, and IgA levels. Body weight gradually decreased in animals fed 4% protein diet (LP) from the third day of diet feeding (*p* < 0.0005) (Figure 1A). At 21 days of low protein intake, LP animals exhibited ~28% body weight loss compared to control (CP) animals that were fed a 14% protein diet (Figure 1A), which represents mild-to-moderate malnutrition status. Leptin serum levels were significantly reduced in malnourished animals (LP and LPi) at 21 days of low protein diet intake when compared to well-nourished groups (CP and CPi) (*p* < 0.0005, Figure 1B). Likewise, IGF-1 serum levels from malnourished mice were significantly diminished when compared to CP and CPi mice after 21 days of diet (*p* < 0.0001, Figure 1C). In contrast, systemic levels of corticosterone and IgA were significantly higher in LP and LPi groups than in CP and CPi groups (*p* < 0.0001, Figure 1D). Indeed, serum IgA levels increased ~30-fold compared to the control group (Figure 1E).

Spleen samples obtained from CPi and LPi mice were analyzed for parasite detection and quantification by qPCR. Successful experimental infection was corroborated by kDNA amplification in the spleen of 100% of the animals infected with *L. infantum* (Figure 1F). In addition, in agreement with our previous report, there was a significant increase in the splenic parasite load of LPi mice at 14 dpi (*p* < 0.05, Figure 1F).

### 3.2. Infection with L. infantum Induced Structural and Inflammatory Alterations in the Duodenum of BALB/c Mice and Such Changes Were Aggravated by Preceding Malnutrition

To determine whether protein malnutrition could alter the architecture of the different sections of the gut, we performed a histopathological analysis of the duodenum, jejunum, ileum, and colon of six mice per group. The length and weight of small and large intestine were similar between the animals of experimental groups (data not shown). The histopathological evaluation revealed different immune and structural alterations—e.g., inflammatory lymphocytic infiltrates, neutrophil, macrophage, or plasma cell infiltrates, lymphoid hyperplasia, villus atrophy, and ulcers (Figure 2 and Figure 3 and Table 1). The frequency of animals undergoing histological alterations showed that at least one animal of each group had some type of perturbation. The CP group presented with the lowest number of alterations per animal and tissue among all groups. In contrast, malnourished and infected animals (LPi) showed a higher frequency of alterations, and those changes were more severe than in the other experimental groups (Table 1).

The most frequent types of alteration induced by malnutrition or infection were lymphocytic or lymphoplasmacytic inflammatory infiltrates of the lamina propria (Table 1). The duodenum and jejunum of 50% of the LP animals presented mild lymphocytic or lymphoplasmacytic infiltrates at the lamina propria; such alteration was also observed in the ileum of 33% LP animals and in the colon—with transmural classification—of 17% of LP mice. 

We observed that infection alone induced alterations in lamina propria and, rarely, in the submucosa of the gut sections of well-nourished animals. Indeed, 50% of animals of the CPi group showed a duodenum with mild lymphocytic or lymphoplasmacytic infiltrates, and 17% with a moderate to severe degree. In the jejunum, 33% of CPi animals exhibited mild lymphocytic or lymphoplasmacytic infiltrates, and 17% moderate to severe lymphocytic inflammatory infiltrates (Table 1). In the ileum and colon, 17% of CPi animals also showed this type of alteration with a mild degree of severity. In addition, neutrophil, macrophage or plasma cell infiltrates were observed in CPi mice in the duodenum (33% of animals), in the jejunum (50% of animals), and in the ileum (17% of animals) (Table 1). 

Remarkably, the frequency of moderate-to-intense inflammatory infiltrates in the lamina propria of at least one segment of gut were higher in the LPi group (50% of animals) than in the CP (17% of animals), CPi (17% of animals) or LP (17% of animals) groups. Furthermore, LPi mice showed a higher frequency of lymphocytic or lymphoplasmacytic infiltrates in the duodenum (83%), jejunum (83%), ileum (50%) and colon (50%) than mice in any of the other experimental groups. In the CP, CPi and LP groups, the frequencies of lymphocytic or lymphoplasmacytic infiltrates ranged from 34% to 67% in the duodenum, 33 to 50% in the jejunum, 17 to 33% in the ileum, and 0 to 33% in the colon (Table 1). 

This analysis also allowed us to determine which section of the gut was more affected by a given variable. The duodenum was the region that suffered the most from the combination of malnutrition and infection; it was followed by the jejunum and ileum (Table 1). In contrast, the colon was clearly the most affected by malnutrition (Table 1).

To obtain a quantitative (and more intuitive) dimension of the qualitative alterations observed in the intestine, we assigned a score to each type of alteration (Appendix A). We observed that each variable (malnutrition or infection) independently induced the occurrence of alterations with moderate magnitudes in the intestine, totalizing a score of 21 for each one of the LP and CPi groups (Figure 4A,B). Remarkably, LPi mice showed the highest score among the experimental groups, displaying a higher number of alterations and a higher degree of severity. Their score totaled 42, which suggested an additive effect through malnutrition and *L. infantum* infection in the frequency and severity of histological alterations in LPi animals (Figure 4A,B). 

Despite the fact that the course of infection in our model was very short (only 14 days), the early structural changes observed in the gut of animals infected with *L. infantum* prompted us to investigate the presence of parasites in different regions. No amastigote forms along the small intestine (duodenum, jejunum, ileum) or colon were detected by immunohistochemistry or qPCR in any of the analyzed sections of the gut from infected mice (CPi and LPi) (data not shown). 

### 3.3. Malnutrition Upregulated mRNA Levels of il-17a, tgfβ, and il-10 in the Duodenum of BALB/c Mice 

As the histopathological analysis showed that the duodenum was severely affected in our model, we decided to assess whether malnutrition and/or infection could alter cytokine gene expression levels in this region. We observed that cytokine mRNA levels were mainly altered by malnutrition. We observed significant increased transcript levels of *tgfβ* (*p* < 0.05) and *il-10* (*p* < 0.05) in LP and LPi animals. In addition, we observed significant increase of *il-12* (*p* < 0.01) in those animals (Figure 5). Noticeably, LP and LPi groups exhibited a significant increase in *il-17A* expression in the duodenum as a result of malnutrition (*p* < 0.03) (Figure 5). The expression of *tnfα* and *ifnγ* showed similar levels among the analyzed groups. 

### 3.4. Infection with L. infantum Induced a Significant Decrease in TNFα, IL-12 and TGFβ Protein Levels in the Duodenum of BALB/c Mice, Whereas Malnutrition Increased the Concentration of Th17-Related Cytokines 

We also evaluated whether malnutrition and/or infection could alter cytokine abundance at the protein level. For this, the concentration of cytokines was measured in the luminal fluid of the duodenum by flow cytometry-based multiplex assay. We observed that the proinflammatory cytokines TNFα and IL-12 were significantly decreased (*p* < 0.01) in the duodenum of infected animals (CPi and LPi) due to infection with *L. infantum* (Figure 6). Moreover, the regulatory cytokine TGFβ was also significantly diminished in the duodena of infected mice (CPi and LPi) (*p* < 0.05) (Figure 6). The duodenal levels of IFNγ and IL-13 did not show significant differences among the groups.

Conspicuously, duodena from malnourished mice (LP and LPi) showed increased levels of cytokines involved with Th17 immune response. Protein levels of IL-17A (*p* < 0.05), IL-21 (*p* < 0.05), and IL-22 (*p* < 0.05) were significantly increased in the LP and LPi mice due to malnutrition (Figure 7). The concentration of duodenal IL-23 did not show significant differences among the experimental groups. Finally, IL-7 was undetectable in all experimental groups (data not shown). Thus, our results showed that the duodena of malnourished-infected mice (LPi) suffered simultaneous decreases of regulatory (TGFβ) and Th1-related cytokines (caused by infection) and increases of Th17 cytokines (caused by malnutrition). 

### 3.5. Whereas CCL5 Protein Levels Were Significantly Increased in the Duodenum of Malnourished-Infected Mice, CXCL10 Was Decreased in These Animals 

The flow cytometry-based multiplex assay also revealed that the concentration of chemokines involved in inflammatory responses and cellular recruitment was significantly altered by malnutrition or by the interaction of both variables (malnutrition and infection). The concentration of CCL5 was significantly increased in the duodenal fluid of LP and LPi animals due to the low protein diet (*p* = 0.01) (Figure 8). In contrast, the concentration of CXCL10 in the duodenal luminal fluid from LPi mice was significantly decreased as a result of the interaction of both variables (malnutrition and infection) (*p* < 0.05). In addition, this chemokine was also significantly diminished in well-nourished infected animals (CPi) due to *L. infantum* infection (*p* < 0.01) (Figure 8). We did not observe significant changes in the abundance of CXCL9 among the experimental groups.

### 3.6. L. infantum Infection Induced a Significant Increase in sIgA Levels in the Duodenum of Well-Nourished Mice, but Malnourished-Infected Animals Exhibited Impaired Secretion of IgA 

Because IgA is the main immunoglobulin present in the secretions that protect mucosal surfaces, we hypothesized that its abundance could be altered due to the histopathological alterations observed in the duodenum. To test this, we measured the concentration of sIgA secreted in the duodenal lumen. The IgA levels were significantly increased in the duodena of infected animals (CPi) (*p* < 0.001) due to *L. infantum* (Figure 9). In addition, duodenal levels of IgA were also increased in LPi mice as a result of interactions between infection and malnutrition (*p* < 0.0001). However, the increase observed in LPi mice was 3.7-fold less than that observed in CPi animals (Figure 9). In agreement with this result, immunofluorescence analysis revealed that infection with *L. infantum* induced a significant increase in the amount of IgA^+^ area in the duodena of CPi animals—it also increased IgA^+^ in LPi mice, but less than was observed in the CPi group (Appendix A). 

## 4. Discussion

Protein malnutrition is a factor that compromises immunity, increasing susceptibility to infectious diseases. In previous work, we demonstrated that protein malnutrition and *L. infantum* infection result in a significant increase in splenic parasite load, a drastic alteration of thymic and splenic microarchitecture, and an impaired cell-mediated immune response [8,9,10,11]. Although malnutrition is a risk factor for VL, little is known about its impact on the architecture of the intestinal mucosae and on intestinal-mediated immune responses to the parasite in animals infected with *L. infantum*.

The intestine is an endocrine organ that plays a crucial role in nutrient response and mediates systemic immune responses [19]. Herein, we studied the effects of protein malnutrition in the gut mucosae of BALB/c mice infected with *L. infantum*. In agreement with our previous works, splenic parasite load was significantly increased in LPi animals, despite the short course of infection, corroborating the early visceralization of parasites in malnourished individuals [9,10]. The spleen, as a target organ of *L. infantum*, suffered changes in its microarchitecture and in the regulation of cell-mediated immune responses [20,21,22]. These events resulted in a dysfunctional local microenvironment that encompassed uncontrolled parasite proliferation and spleen scaping, contributing to the systemic deterioration of the infected individual [20,21,22]. In turn, malnutrition per se also induced drastic alterations in splenic organization [8,10]. We previously showed that when malnutrition precedes parasite infection, it alters T cell subpopulations and proinflammatory responses in the spleen, and induces a precocious increase in parasite load in the organ, accelerating and impairing the pathological events of the infection course [8,10]. Such uncontrolled parasite proliferation and scaping may target surrounding tissues that have also been affected by malnutrition, such as the gut mucosae. 

The cases of non-canonical visceralization occur at a much lower frequency and are mainly observed in immunosuppressed individuals, e.g., those coinfected with HIV [23,24,25]. As malnutrition is an immunosuppressive condition, the colonization of the intestine by *L. infantum* may be a probable event. In this study, we did not detect parasites in the intestinal tissues. However, our course of infection was short, (only 14 days) and parasite load in those tissues may have been under the limit of detection of our method. Nevertheless, we observed important histopathological alterations in the gut mucosae of well-nourished mice infected with *L. infantum*—which were exacerbated in malnourished-infected ones. Thus, even without detectable parasites, the significant histopathological alterations observed at 14 dpi in the duodena of CPi and LPi mice indicated that this region of the intestine is sensitive to parasitic infection in the early stages of the *L. infantum* infectious course. The infection of the spleen could be sensed—in a manner of speaking—by the small intestine, resulting in the structural and molecular alterations reported in the duodena here, which were exacerbated when malnutrition preceded infection.

Although the current understanding of the communication between gut mucosae and systemic immune responses is very narrow, such interaction is required for an appropriate host immune response. Exosome release from intestinal epithelial cells is a protective event against the parasite *Cryptosporidium parvum* [26]. Those exosomes can activate splenocytes and induce an inflammatory response by releasing several effector molecules that could be crucial for the systemic host immune responses against that parasite [27]. Similarly, endocrine communication between the infected organs and intestinal tissue could be occurring during malnutrition and *L. infantum* infection. In addition, in our model, malnourished animals presented significant reduction of systemic levels of IGF-1, and leptin—which are recognized markers of malnutrition [9,28,29,30]—whereas those animals exhibited significant increased levels of systemic IgA and corticosterone, composing a compromised endocrine landscape.

High levels of IgA secreted in the sera of malnourished animals were in agreement with levels observed in malnourished children and BALB/c mice elsewhere [31,32,33]. The levels of immunoglobulin decrease observed after the administration of a therapeutic diet in children indicates a direct influence of the diet on IgA levels [31,33]. Serum IgA diminishes the release of pro-inflammatory cytokines by peripheral blood mononuclear cells (PBMCs) [34,35] and the phagocytic activity and chemotaxis of polymorphonuclear leukocytes (PMNs) [36,37,38]. Nevertheless, some reports indicate that IgA activity depends on its titers. These reports postulate that, at low concentrations, IgA activity favors inhibitory immune signals, whereas at high concentrations, it leads to the activation of signals for pathogen clearance [39]. In our model, the elevated concentration of IgA in serum was accompanied by a systemic anti-inflammatory profile in malnourished and infected mice [9], suggesting that, at a systemic level, malnutrition may be favoring inhibitory immune signals. 

Nevertheless, a different picture was presented when regional responses were analyzed. The duodena of infected animals (CPi and LPi) showed increased levels of secreted IgA because of *L. infantum* infection (alone or in interaction with malnutrition). One of the central components of the immune axis in the gut is IgA, which is regularly secreted in mucous membranes and peripheral lymphoid tissues, providing humoral protection against different pathogens, environmental injuries, and dietary components [40,41]. IgA controls organ homeostasis by controlling the proliferation, establishment, and translocation of different microbial populations in the intestine, neutralizing toxins produced by pathogens, and cleaning the waste produced during tissue restoration [42,43]. Thus, the increase in the production and secretion of IgA into the luminal space of the duodenum suggests an early protective immune response related to *L. infantum* infection. The spleen maintains constant communication with the intestine tissue acting as a reservoir of immune cells that can emigrate in response to gut infections. After antigen stimulation, splenic IgM^+^ B cells can migrate to the intestinal lamina propria, where they preferentially differentiate into IgA^+^ plasma cells and proliferate [44]. In line with that evidence, our results indicated that the duodena of well-nourished-infected animals sensed the spleen infection and modulated IgA secretion through an increase in IgA^+^ cells, possibly as a result of an endocrine communication with other lymphoid tissues. Interestingly, animals that suffered malnutrition and were further infected (LPi) were not capable of secreting IgA at the same levels that well-nourished-infected mice (CPi), indicating that the protein-restricted diet affected the IgA-mediated immune response in the small intestines of these animals. 

Secretion of IgA is regulated by microbiota-specific Th17 cells that secrete IL-17A, IL-17F, IL-21 and IL-22 [45]. Recently, it was shown that IL-21 plays a critical role in limiting growth of atypical commensal bacteria via enhancing T cell-dependent IgA responses to them [46]. Interestingly, in our model, increased IgA secretion in the duodenum was accompanied by increased levels of IL-17A. Moreover, we also observed a significant increase of IL-21 and IL-22 in the duodena of malnourished mice. These results suggest that malnourished mice may have suffered dysbiotic events, and that local inflammatory responses occurred in those animals. On one hand, dietary changes can induce significant alterations in the gut microbiome [47,48] and, on the other hand, changes in the microbiome may influence the severity of infectious diseases, as seen for malaria [49]. However, further studies should be done in order to corroborate the relationship between malnutrition, IgA production and potential dysbiotic events in our model.

In homeostatic conditions, IL-17A is recognized for maintaining barrier integrity, protecting against pathogens, and regulating the recruitment of neutrophils to barrier sites [50]. Additionally, IL-17A can act as a proinflammatory cytokine and a potent inductor of the innate immune response [51]. During inflammatory bowel disease (IBD), immune cells upregulate the secretion of IL-17, IL-21, IL-22, and IL-23 cytokines in the intestinal mucosa [52]. The role of IL-17 in IBD is a matter of debate, but most studies postulate that it participates in the intensification of mucosal inflammation [53,54]. IL-22 is also a proinflammatory cytokine and a major regulator of the intestinal barrier; it mediates epithelial regeneration with a protective role in IBD [55]. Despite the beneficial effects of both proteins, it is generally accepted that IL-17 and IL-22 overactivation may cause chronic inflammatory diseases. Noticeably, LP and LPi groups exhibited a significant increase of IL-17A, IL-21, and IL-22 protein levels in the duodenal fluid. In addition, the duodenum of LPi mice was more affected by the infiltration of neutrophils, macrophages, plasma cells, and lymphocytic infiltrates in the lamina propria than the other groups. 

IL-22 can act synergistically with TNFα in response to infection-promoting chemokine expression (including CXCL9, CXCL10 and CCL5) by human keratinocytes [56]. Accordingly, we observed that increased levels of IL-22 were accompanied by significant increasing in CCL5 protein levels in LPi animals, supporting both the localized inflammatory environment and the increased events of cellular infiltrates in the duodena of that experimental group. Interestingly, although CXCL10 was high in LP animals, the increase seemed to be counteracted by *L. infantum* infection, resulting in diminished levels of this cytokine in LPi mice and reinforcing the observation that the duodenum was sensitive to both variables. Overall, our results suggest that a low protein intake increased the Th17 inflammatory response in the small intestine, altering its histological organization and increasing the recruitment of immune cells. Such dietary modulation of immune responses in the gut could shape the severity of *L. infantum* infections as observed in other infectious disease models [49]. Further experiments will be necessary to understand how malnutrition modulates the extraintestinal infection (in the spleen) and whether the modulation of gut immune responses is responsible for the difference in splenic parasite load between well-nourished and malnourished mice. 

## 5. Conclusions

Our results show that malnutrition induced a variety of structural, immunological, and functional changes in the duodena of mice, influencing the way that enteric region sensed the infection with *L. infantum*, and impairing the early gut responses to extraintestinal infection. Increased levels of duodenal IL-17A, IL-21, IL-22, CXCL5 due to malnutrition—together with decreased levels of TNFα, IL-12, TGFβ and CXCL10 and elevated levels of IgA due to infection—revealed that the duodenum was sensitive to both variables and suggested local inflammation in malnourished animals. Those cytokines and chemokines could have been involved in the infiltration of immune cells and histological alterations observed in that region. All of those alterations were observed in concomitance with significantly higher parasite load in the spleen of malnourished mice as compared to well-nourished mice. Thus, our results suggest that infection in the spleen is sensed in different regions of the gut, altering duodenal immune responses, and that these changes are impaired and/or exacerbated when a precondition of malnutrition exists. 

## Figures and Tables

**Figure 1 microorganisms-09-01270-f001:**
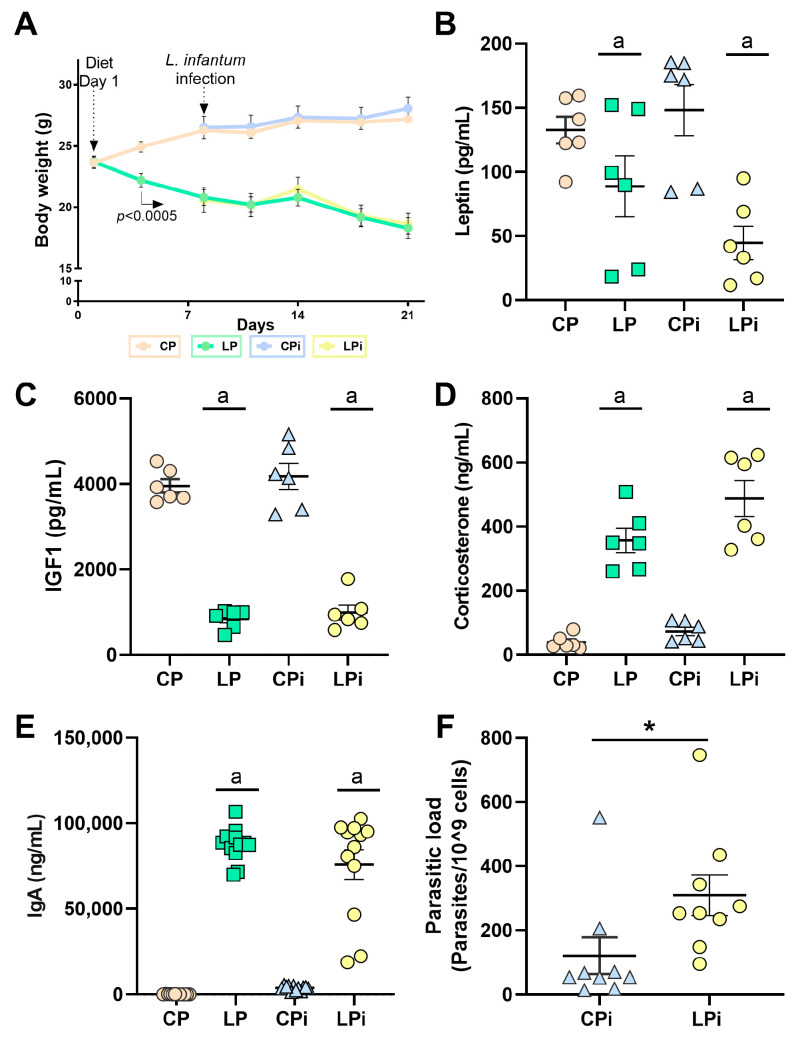
Bodyweight and hormonal levels in malnourished BALB/c mice infected with *L. infantum*. (**A**) Male BALB/c mice were fed a 14% (*n* = 24, control protein (CP)) or 4% (*n* = 24, low protein (LP)) protein diet for 21 days. On day 7 of the experimental period, half of the animals were infected with *L. infantum* and the other half received an injection of saline solution. Bodyweight was recorded every third day and expressed as average ± SEM; *n* = 12 mice in each group. Statistical differences before the day of infection were determined by Student’s *t*-test (*p* < 0.0005). After infection, a two-way ANOVA with Tukey’s *post-hoc* test was used (*p* < 0.0001). (**B**) Leptin, (**C**) total IGF-1 (Insulin-like growth factor 1), (**D**) corticosterone, and (**E**) immunoglobulin A (IgA) serum levels were analyzed in the experimental groups (*n* = 6 per group). Leptin, IGF-1, and IgA were measured by commercial ELISA assays, whereas corticosterone was measured by radioimmunoassay. Statistical differences due to diet (a), infection (b), or (c) interaction between diet and infection were determined by two-way ANOVA (*p* < 0.05). (**F**) Parasite load was determined by qPCR in the spleen and statistical differences were determined by Student’s *t*-test (* *p* < 0.05). CP: control protein, animals fed 14% protein diet; LP: low protein, animals fed 4% protein diet, CPi: animals fed 14% protein diet and infected; LPi: animals fed 4% protein diet and infected.

**Figure 2 microorganisms-09-01270-f002:**
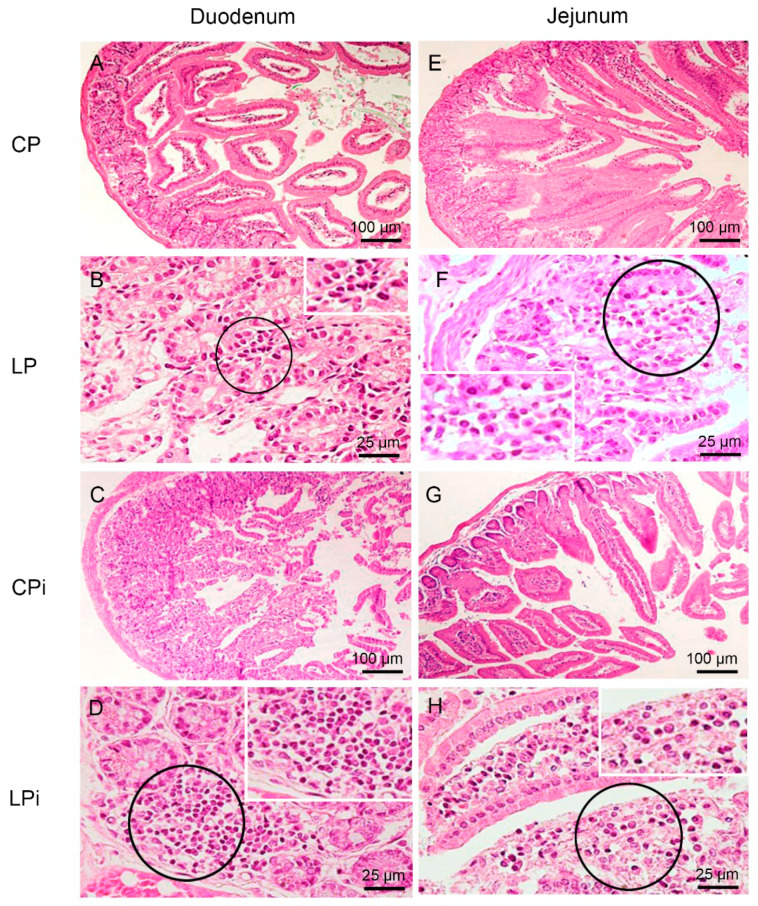
Histopathological analysis of duodenum and jejunum from BALB/c mice. CP: animals fed 14% protein diet, LP: animals fed 4% protein diet, CPi: animals fed 14% protein diet and infected, LPi: animals fed 4% protein diet and infected. (**A**–**D**) Duodenum. H&E. (**A**) CP mouse without histological alterations. (**B**) LP mouse showing mild and focal lymphocytic infiltrate (circle and inset) in the lamina propria. (**C**) CPi mouse showing moderate to severe and diffuse lymphocytic infiltrate in the lamina propria. (**D**) LPi mouse showing moderate to severe and diffuse lymphoplasmacytic infiltrate (circle and inset). (**E**–**H**) Jejunum. H&E (**E**) CP mouse without histological alterations. (**F**) LP mouse showing mild and focal lymphocytic infiltrate (circle and inset) in the lamina propria. (**G**) CPi mouse without histological alterations. (**H**) LPi mouse showing moderate to severe lymphoplasmacytic infiltrate (circle and inset) in the lamina propria.

**Figure 3 microorganisms-09-01270-f003:**
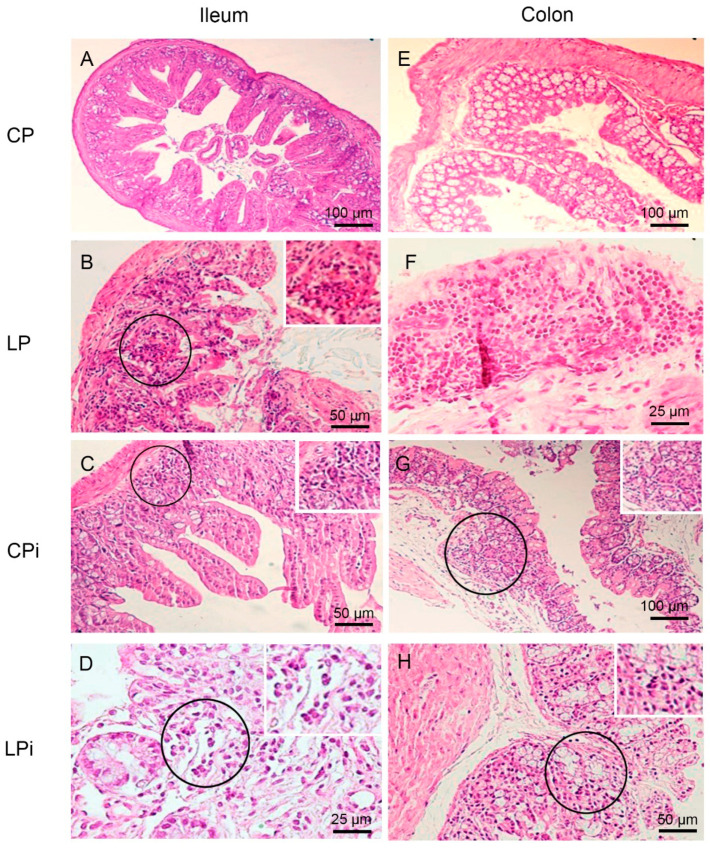
Histopathological analysis of ileum and colon from BALB/c mice. CP: animals fed 14% protein diet, LP: animals fed 4% protein diet, CPi: animals fed 14% protein diet and infected, LPi: animals fed 4% protein diet and infected. (A-D) Ileum. H&E. (**A**) CP mouse without histological alterations. (**B**) LP mouse showing mild and diffuse lymphocytic infiltrate (circle and inset) in the lamina propria. (**C**) CPi mouse showing mild and diffuse lymphoplasmacytic infiltrate (circle and inset) in the lamina propria. (**D**) LPi mouse showing moderate to severe and diffuse lymphoplasmacytic infiltrate (circle and inset) in the lamina propria. (**E**–**H**) Colon. H&E (**E**) CP mouse without histological alterations. (**F**) LP mouse showing moderate to severe and diffuse lymphocytic infiltrate in the lamina propria. (**G**) CPi mouse showing mild and diffuse lymphocytic infiltrate (circle and inset) in the lamina propria. (**H**) LPi mouse showing mild and diffuse lymphoplasmacytic infiltrate (circle and inset) in the lamina propria.

**Figure 4 microorganisms-09-01270-f004:**
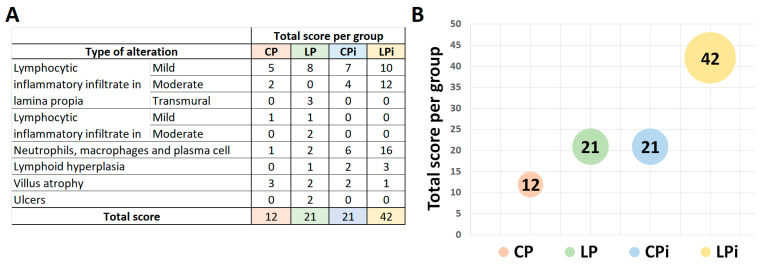
Total score of histopathological changes observed in the intestine regions of malnourished BALB/c mice infected with *L. infantum*. (**A**) The score values represent the sum of the changes observed in each experimental group according to the presence or absence of the alteration and its degree of severity (Appendix A). (**B**) Total score of immunological and structural changes observed in each experimental group. *n* = 6 animals per group. CP: animals fed 14% protein diet, LP: animals fed 4% protein diet, CPi: animals fed 14% protein diet and infected, LPi: animals fed 4% protein diet and infected.

**Figure 5 microorganisms-09-01270-f005:**
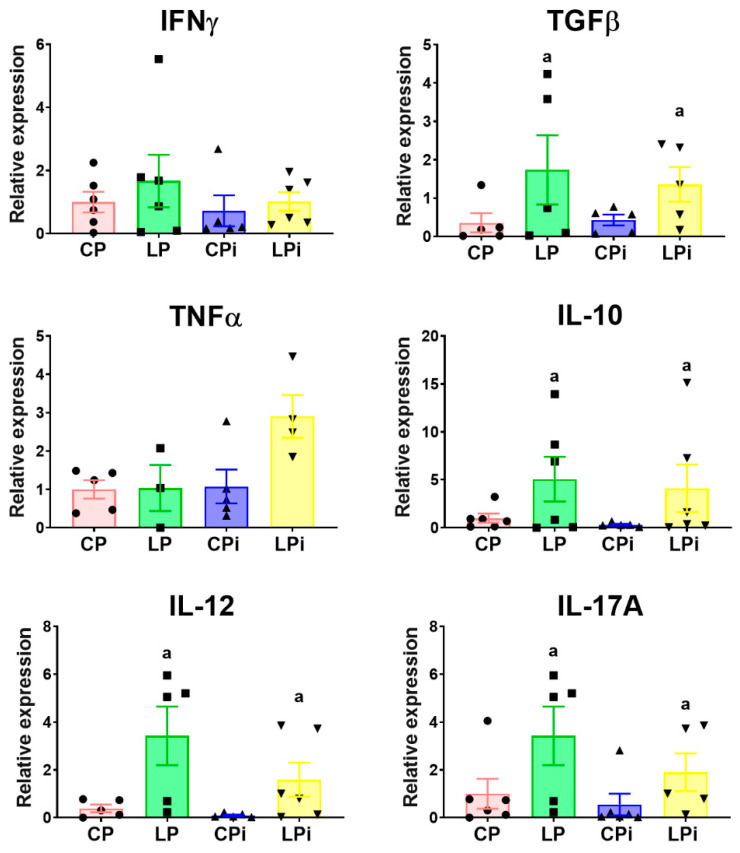
Cytokine expression in the duodenum from malnourished BALB/c mice infected with *L. infantum. il-12, ifnγ, tnfα, tgfβ* and *il-17* mRNA expression levels measured by qPCR in the duodenum of each experimental group. The values are expressed as normalized ratios between the target gene expression and the geometric median of the *ATP-5, GAPDH* and *CYC-1* genes. Statistical differences due to (a) diet (*p* < 0.001) were determined by two-way ANOVA. CP: animals fed 14% protein diet; LP: animals fed 4% protein diet, CPi: animals fed 14% protein diet and infected; LPi: animals fed 4% protein diet and infected.

**Figure 6 microorganisms-09-01270-f006:**
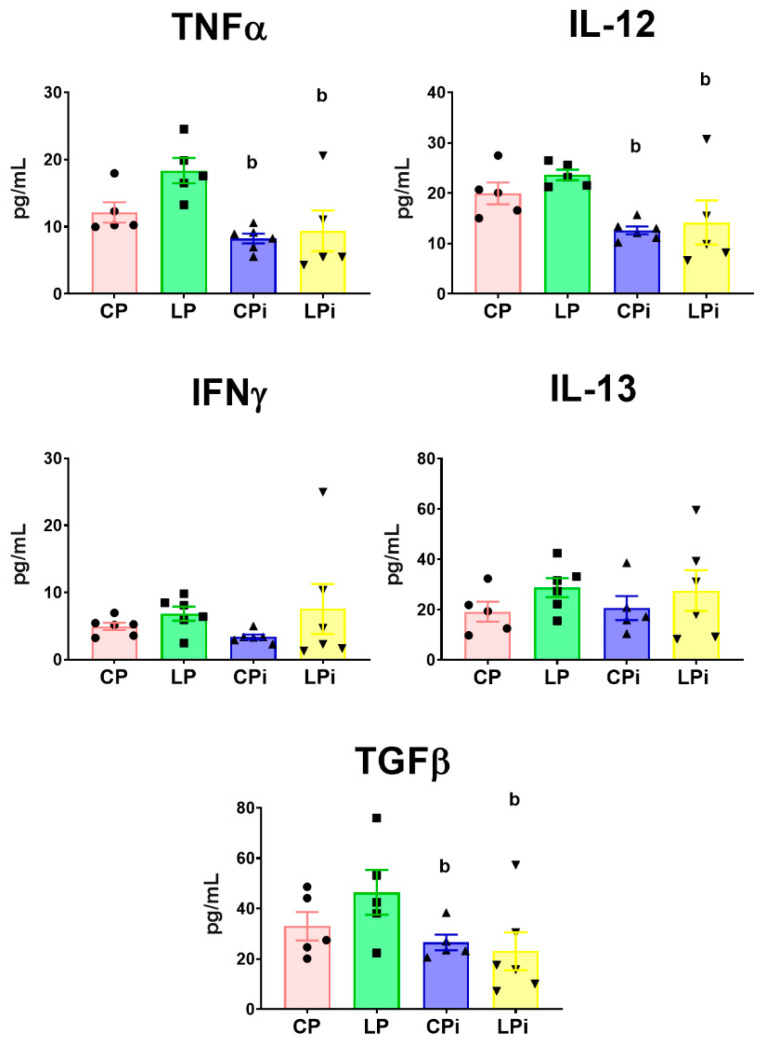
Cytokines levels in the luminal fluid of duodenum from malnourished BALB/c mice infected with *L. infantum***.** TNFα, IL-12, IFNγ, IL-13 and TGFβ protein levels were measured in the luminal fluid of the duodenum at 14 days post-infection using a flow cytometry-based multiplex assay. Significant differences due to infection (b), *p* < 0.01 for TNFα and IL-12 and *p* < 0.05 for TGFβ, as determined by two-way ANOVA. CP: animals fed 14% protein diet; LP: animals fed 4% protein diet, CPi: animals fed 14% protein diet and infected; LPi: animals fed 4% protein diet and infected.

**Figure 7 microorganisms-09-01270-f007:**
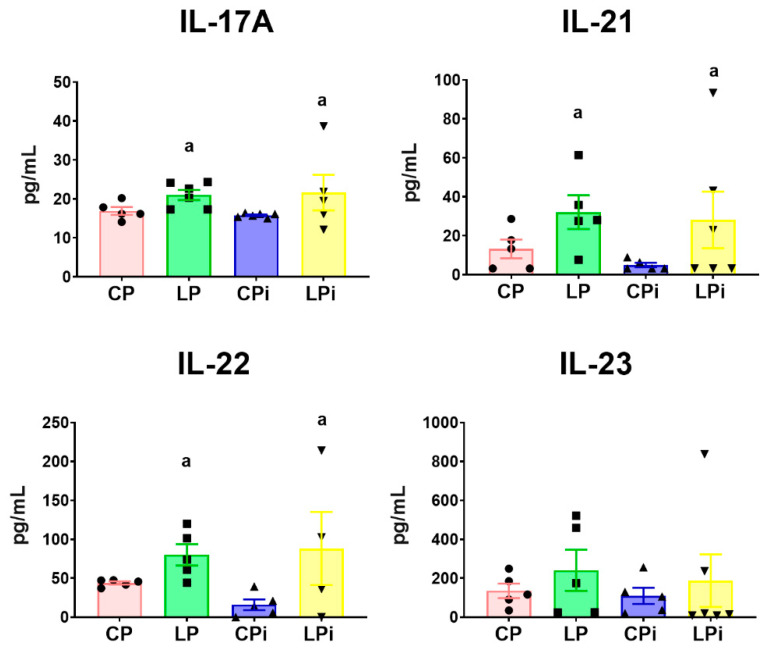
Protein levels of Th-17 related cytokines in the luminal fluid of duodena from malnourished BALB/c mice infected with *L. infantum*. IL-17A, IL-21, IL-22, and IL-23 protein levels in the luminal fluid of duodena were measured at 14 days post-infection using a flow cytometry-based multiplex assay. a: significant differences due to diet (*p* < 0.05) determined by two-way ANOVA. CP: animals fed 14% protein diet; LP: animals fed 4% protein diet, CPi: animals fed 14% protein diet and infected; LPi: animals fed 4% protein diet and infected.

**Figure 8 microorganisms-09-01270-f008:**
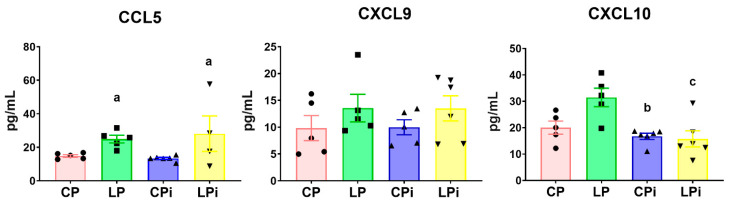
Chemokine levels in the luminal fluid of duodenum from malnourished BALB/c mice infected with *L. infantum*. CCL5, CXCL9, and CXCL10 levels in the luminal fluid of duodena were measured at 14 days post-infection using a flow cytometry-based multiplex assay. Significant differences due to diet (a), *p* < 0.05; significant differences due to infection (b), *p* < 0.01; significant differences due to interaction between diet and infection (c), *p* < 0.05, as determined by two-way ANOVA. CP: animals fed 14% protein diet; LP: animals fed 4% protein diet; CPi: animals fed 14% protein diet and infected; LPi: animals fed 4% protein diet and infected.

**Figure 9 microorganisms-09-01270-f009:**
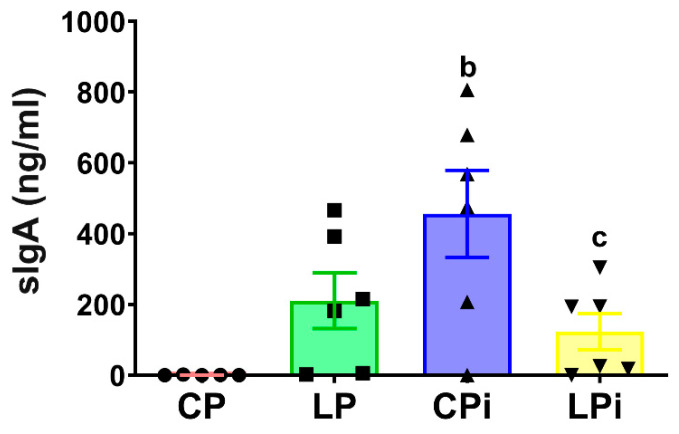
Levels of secreted IgA in duodenum of malnourished BALB/c mice infected with *L. infantum*. The levels of secreted IgA measured by ELISA according to the protocol described in materials and methods. Significant differences due to infection (b), *p* < 0.05; significant differences due to interaction between diet and infection (c), *p* < 0.01. CP: animals fed 14% protein diet; LP: animals fed 4% protein diet, CPi: animals fed 14% protein diet and infected; LPi: animals fed 4% protein diet and infected.

**Table 1 microorganisms-09-01270-t001:** Frequency of histological changes observed in the intestine of malnourished BALB/c mice infected with *L. infantum* *.

Type of Alteration		CP				LP				CPi				LPi			
		D	J	I	C	D	J	I	C	D	J	I	C	D	J	I	C
**Lymphocytic inflammatory infiltrate in lamina propria**	**Mild**	17% (1/6)	17% (1/6)	17% (1/6)	0/6	50% (3/6)	50% (3/6)	33% (2/6)	0/6	50% (3/6)	33% (2/6)	17% (1/6)	17% (1/6)	50% (3/6)	50% (3/6)	33% (2/6)	33% (2/6)
	**Moderate**	17% (1/6)	0/6	0/6	0/6	0/6	0/6	0/6	0/6	17% (1/6)	17% (1/6)	0/6	0/6	33% (2/6)	33% (2/6)	17% (1/6)	17% (1/6)
	**Transmural**	0/6	0/6	0/6	0/6	0/6	0/6	0/6	17% (1/6)	0/6	0/6	0/6	0/6	0/6	0/6	0/6	0/6
**Lymphocytic inflammatory infiltrate in submucose (SM)**	**Mild**	0/6	0/6	17% (1/6)	0/6	0/6	0/6	0/6	17% (1/6)	0/6	0/6	0/6	0/6	0/6	0/6	0/6	0/6
	**Moderate**	0/6	0/6	0/6	0/6	0/6	0/6	0/6	17% (1/6)	0/6	0/6	0/6	0/6	0/6	0/6	0/6	0/6
**Neutrophils, macrophages and plasma cells infiltrate**		0/6	0/6	17% (1/6)	0/6	17% (1/6)	17% (1/6)	0/6	0/6	33% (2/6)	50% (3/6)	17% (1/6)	0/6	83% (5/6)	83% (5/6)	50% (3/6)	50% (3/6)
**Lymphoid hyperplasia**		0/6	0/6	0/6	0/6	0/6	0/6	0/6	17% (1/6)	0/6	33% (2/6)	0/6	0/6	17% (1/6)	0/6	17% (1/6)	17% (1/6)
**Villus atrophy**		0/6	33% (2/6)	17% (1/6)	0/6	17% (1/6)	0/6	17% (1/6)	0/6	0/6	17% (1/6)	17% (1/6)	0/6	0/6	0/6	17% (1/6)	0/6
**Ulcers**		0/6	0/6	0/6	0/6	0/6	0/6	0/6	33% (2/6)	0/6	0/6	0/6	0/6	0/6	0/6	0/6	0/6

***** Frequency is denoted by the percentage of animals presenting the alteration; parentheses denote the number of animals presenting the alteration over the total animals analyzed (*n* = 6, for each experimental group). The lymphocytic inflammatory infiltrates of the lamina propria and submucosa (SM) are classified according to the severity of the change (mild, moderate, or transmural). Colors highlight the alterations observed in each tissue of each experimental group. CP: animals fed 14% protein diet (control); LP: animals fed 4% protein diet; CPi: animals fed 14% protein diet and infected with *L. infantum*; LPi: animals fed 4% protein diet and infected with *L. infantum*. D: duodenum; J: jejunum; I: ileum; C: colon. *n* = 6 animals per group.

## Data Availability

The data presented in this study are available in Figure 1, Figure 2, Figure 3, Figure 4, Figure 5, Figure 6, Figure 7, Figure 8 and Figure 9 and Appendix A.

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
