# Peer review of "Malnutrition Aggravates Alterations Observed in the Gut Structure and Immune Response of Mice Infected with Leishmania infantum"

_microorganisms, 2021, doi:10.3390/microorganisms9061270_

Round 1

Reviewer 1 Report

The manuscript entitled “Malnutrition aggravates alterations observed in the gut structure and immune response of mice infected with Leishmania infantum” presents novel and interesting studies to be published in the journal Microorganisms. The manuscript is well-written, the data are appropriately presented and the introduction gives the essential background.

However, some comments have to be adrressed:

  1. The study is carried out 2 weeks p.i., what happens at longer times of infection? If parasite burden in spleens is higher, perhaps parasites are found in intestine and changes at different levels are higher. Do you have some results about this?
  2. In your murine model, at 2 weeks p.i., T cell subpopulations in the spleen and intestine are altered?

Reviewer 2 Report

The work of Caitan-Albarracin et al convincingly demonstrates that malnutrition exacerbates the immune response (and structural integrity of the gut) of mice infected with L. infantum. Overall, I liked the manuscript and find its conclusions justified (even though, I think the main results are expected). 

A few minor and cosmetic comments:

ln: 87:  was this Schneider's Drosophila medium? Please specify its source. 

Methods section: A) please delete catalog numbers and "Inc." B) Please double-check the origin of all the reagents and equipment - f.e. I could not find the company name "NanoDrop Techlnologies". C) Please be consistent in your description (Name, Country). 

Please revise figure legends - they should be self-explanatory. F.e. In Fig. 1 you should explain what is what again: CP, LP, CPi, LPi (especuially because they are labeled differently - this also needs to be specified).   
